# Molecular Mechanisms of Neutrophil Extracellular Trap (NETs) Degradation

**DOI:** 10.3390/ijms24054896

**Published:** 2023-03-03

**Authors:** Urszula Demkow

**Affiliations:** Department of Laboratory Diagnostics and Clinical Immunology of Developmental Age, Medical University of Warsaw, 02-091 Warszawa, Poland; urszula.demkow@uckwum.pl

**Keywords:** neutrophil extracellular traps, degradation, DNAses, macrophages, autoimmunity, thrombosis, cancer

## Abstract

Although many studies have been exploring the mechanisms driving NETs formation, much less attention has been paid to the degradation and elimination of these structures. The NETs clearance and the effective removal of extracellular DNA, enzymatic proteins (neutrophil elastase, proteinase 3, myeloperoxidase) or histones are necessary to maintain tissue homeostasis, to prevent inflammation and to avoid the presentation of self-antigens. The persistence and overabundance of DNA fibers in the circulation and tissues may have dramatic consequences for a host leading to the development of various systemic and local damage. NETs are cleaved by a concerted action of extracellular and secreted deoxyribonucleases (DNases) followed by intracellular degradation by macrophages. NETs accumulation depends on the ability of DNase I and DNAse II to hydrolyze DNA. Furthermore, the macrophages actively engulf NETs and this event is facilitated by the preprocessing of NETs by DNase I. The purpose of this review is to present and discuss the current knowledge about the mechanisms of NETs degradation and its role in the pathogenesis of thrombosis, autoimmune diseases, cancer and severe infections, as well as to discuss the possibilities for potential therapeutic interventions. Several anti-NETs approaches had therapeutic effects in animal models of cancer and autoimmune diseases; nevertheless, the development of new drugs for patients needs further study for an effective development of clinical compounds that are able to target NETs.

## 1. NETs Generation and Their Biological Role

Neutrophil extracellular traps (NETs) are web-like structures built from chromatin fibers decorated with antimicrobial enzymes and histones, serving as a trap, immobilizing and killing microorganisms, and therefore limiting their spread [1]. These structures, crucial for the proper functioning of the immune system, were described in 2004 by Arturo Zychlinsky and his group [1]. The clinical relevance and high impact of this phenomenon on thousands of physiological and pathological processes confirm that this is a Nobel prize-worthy discovery. As we previously described in detail, NETs release by neutrophils occurs primarily through a cell death mechanism termed NETosis [2]. This process begins with neutrophil activation by microbial products including endotoxins, with the support of platelets, proinflammatory cytokines or other danger signals. Several molecules bind to neutrophil receptors (such as Toll-like receptors (TLRs) and complement receptors) to activate neutrophils and trigger NETosis. Various factors of microbial origin, such as the size of microorganism and ROS, are regulators of NETosis—further reviewed in [2,3]. A next step is the activation of nicotinamide-adenine-dinucleotide-phosphate (NADPH) oxidase and intracellular granular proteases, followed by histones citrullination and chromatin decondensation [1]. Citrullination is mediated by the enzyme peptidylarginine deiminase 4 (PAD4), which removes positive charges from core histones by converting arginine residues to citrullines, thereby weakening the interaction between histones and DNA [1,2]. Next, the nuclear envelope of neutrophils breaks, and decondensed nuclear chromatin is expulsed, mixing with cytoplasmic and granule components, including neutrophil elastase, myeloperoxidase, cathelicidin antimicrobial peptide (LL37), high mobility group protein B1 (HMGPB1) and cathepsin G and proteinase 3 [1,2]. Subsequently, the cell membrane permeabilizes, NETs are released out of the cell and neutrophil dies in the process of NETosis [1,2,3]. An alternative mechanism maintaining structural integrity of granulocyte, referred as non-suicidal NETosis, starts from the blebbing of the nuclear envelope and fast exportation of microvesicles containing NETs outside of the cell in the absence of nuclear destruction. This is a fast reaction of the infiltrating granulocytes attracted to the sites of infection [1,2]. The neutrophils stay alive and retain the ability to combat bacteria after expulsion of their DNA. As a third mechanism, NETs can be generated from mitochondrial DNA, previously described by our group in [2,3]. In this process, mitochondria move to the cell surface and expel NETs. NETs are important contributors to the neutrophil antimicrobial response in tissues and vessels. NETs release and their degradation by DNases must be tightly regulated to prevent excessive inflammatory reactions [1,2]. Both the overproduction and the defects of the NETs clearance were found to promote numerous pathologies [2,3]. Neutrophils die at inflamed tissues undergoing netosis or one of the various other cell death mechanisms as apoptosis, necrosis, necroptosis, pyroptosis or autophagy [1,2,3,4]. The interplay of these processes is necessary for combatting foreign invaders and accounts for a further resolution of the inflammation [2,3]. All cell death mechanisms contribute to the regulation of neutrophils number, but also guaranties the degradation of their cargo and regulate the production of pro- and anti-inflammatory mediators [2,5]. The dysregulation of neutrophil death occurs in various pathological conditions such as sepsis and ARDS [3,5]. It is assumed that netosis, apoptosis and autophagy sharply rely on NADPH oxidase function and ROS production [2,3]. The redox disbalance within neutrophil likely accelerates the induction of the death machinery [2]. Following a stimulus, the generation of ROS is a prerequisite for autophagic, apoptotic or netotic processes and can be activated as a consequence of elevated ROS levels [2,3,4,5]. Moreover, stimulation of neutrophils with Toll-like receptor (TLR) agonists, PMA or phagocytosis of microorganisms activates netotic machinery or other type of cell death [1,2,3]. Further study is needed to better understand the molecular mechanisms regulating neutrophil death decisions. In particular, efforts should be made to gain insight into the starting points and upstream events of neutrophil netosis, autophagy, apoptosis or simply necrosis.

## 2. NETs in Clinical Pathology

NETs not only play a key role as a host defense mechanism against local and systemic infections, but if overproduced and persistent, exacerbate acute and chronic infectious diseases and participate in a variety of non-infectious conditions, all reviewed in [1,2,3,4,5].

NETs formation has been linked to an extraordinarily broad range of biological events. Netting neutrophils have the capacity to actively participate in multiple cellular and molecular cascades by releasing the cargo of mediators, including histones, metalloproteinases, cytokines, free DNA, proteases and ROS [1,2]. The NETs contribute to an overactivation of immune cells, the generation of thrombi in the circulation, endothelial and epithelial cells damage, vascular and bronchial occlusion, local tissue destruction, amplification of the vicious circle of the inflammatory response, etc., all processes are discussed in detail in our previous reviews [3,4,5,6]. The NETs interact with dendritic cells and macrophages, which in turn release interleukin 1 β (IL-1β) and interferon α (IFNα). NETs can also activate T-cell to release IFNα and IFNγ, deeply discussed in [3]. DNA decorated with histones and proteases, by disturbing homeostasis of the immune system, is involved in the pathogenesis of various inflammatory diseases such as psoriasis, rheumatoid arthritis, granulomatosis with polyangiitis, systemic lupus erythematosus (SLE), preeclampsia, cystic fibrosis, chronic otitis media, atherosclerosis, stroke, pancreatitis or severe COVID-19 [4,6,7]. NETs are also implicated in various non-inflammatory pathological processes, such as coagulation disorders, cancer, diabetes and wound healing [4,6,8].

### 2.1. NETs in Autoimmune Diseases

As soon as the very first publications describing the NETs appeared, it was recognized that this structure is a potent source of various autoantigens which may induce autoimmune reaction and contribute to the development of autoimmune diseases [2,9]. Moreover, NETs components may act as damage-associated molecular patterns (DAMPs), and opposite DAMPs are able to induce NETs formation, generating a vicious circle of inflammation exaggerating organ damage and causing remote organ injury in the course of chronic inflammatory processes [10]. NETs have been implicated in numerous autoimmune disorders, including both systemic and local diseases, which may affect different organs (kidneys, joints, skin, blood vessels, lungs, central and peripheral nervous system) [2,6,9]. The accumulation of NETs and its components in the circulation correlates with the formation of anti-double-stranded DNA (dsDNA), anti-nucleosomes and anti-histones antibodies being considered a pathogenic factor for SLE [11]. The immune complexes built from these materials and immunoglobulins may depose in the glomeruli and cause lupus nephritis (LN). NETs are also engaged in the pathological processes in anti-neutrophil cytoplasmic antibodies (ANCA)—associated vasculitis, psoriasis and gout [2,9]. Elevated levels of circulating NETs markers were observed in multiple sclerosis [6]. In addition, elevated NETs components were found in peripheral blood, synovial fluid, rheumatoid nodules and skin of rheumatoid arthritis patients, and the NETs markers were positively associated with the concentration of anti-citrullinated protein antibodies (ACPA) [9]. Moreover, a majority of monoclonal antibodies found in synovial fluid and serum from rheumatoid arthritis patients reacts with citrullinated proteins (histones H2A/H2B, fibrinogen and vimentin) [11]. The citrullination of various proteins is a prominent feature of rheumatoid arthritis but, concomitantly, it plays an important role in the process of NETs formation [12]. The single nucleotide polymorphism in the gene encoding a protein tyrosine phosphatase (PTPN22) at position 1858 resulting in a missense mutation that converts an arginine a tryptophan was strongly associated with rheumatoid arthritis and excessive citrullination [13]. Chang et al. confirmed that the modification of C1858T disrupted the interaction between PTPN22 PAD4, followed by enhanced citrullination and exuberant NETs formation [13]. As described above, the citrullination of histones by PAD4 and the activation of the Raf-MEK-ERK signaling pathway have been described as necessary for their respective effects on histone degradation and expression of antiapoptotic pathways, subsequently leading to the release of decondensed chromatin DNA [1,2,13].

### 2.2. NETs—Coagulation

All components of NETs (DNA, histones and proteases) display procoagulant properties in the vascular compartment and in the surrounding tissues [5,8]. NETs promote venous, arterial and microvessels thrombosis by activating platelet adhesion and aggregation, providing a physical scaffold for thrombus formation from platelets and fibrin and being a trap for erythrocytes, all occluding the capillaries [8]. Remnants of NETs (dsDNA, myeloperoxidase-DNA complexes and citrullinated histones) activate coagulation cascade by increasing the protease activity of coagulation factors including thrombin [14]. NETs—derived dsDNA—directly activate the extrinsic pathway of coagulation, while NETs remnants promote thrombosis by the induction of tissue factor release from activated platelets and monocytes to initiate the intrinsic pathway as described in detail in [8]. Histones impair the function of coagulation inhibitors including thrombomodulin, thus promoting thrombin generation [15]. NETs are also required for the propagation of thrombi by binding and activating factor XII [5,8]. Neutrophil elastase promote coagulation by inactivating tissue factor pathway inhibitors, thus further increasing coagulation and fibrin deposition in vivo [8]. NETs aggregates can also occlude other tubular structures such as the bile and pancreatic ducts, provoking alterations of organ function and inflammation known as neutrophil extracellular trap-driven occlusive diseases [16].

### 2.3. NETs in COVID-19 and in Other Severe Infections

Neutrophils and their products, including NETs, strongly contribute to acute lung injury, multi-organ damage and mortality in COVID-19, as reviewed by Szturmowicz and Demkow [5]. The markers of NETs formation, such as circulating DNA, nucleosomes, citrullinated histones, neutrophil elastase activity or myeloperoxidase-DNA complexes were found in sera of COVID-19 patients at a higher level as compared to healthy donors [5,16]. Moreover, the concentration of those markers significantly decreased in the recovery phase of COVID-19 [5,16]. Endothelial and pulmonary alveoli epithelial cell injury, as well as the disruption of alveolar-capillary barrier, a hallmark of severe pulmonary COVID-19, have been reported to be caused by NETs and their components [5,17]. The DNA threads form large conglomerates causing local obstruction of the small bronchi, and together with neutrophil elastase, are responsible for the overproduction of mucus by goblet cells of surface epithelia [5]. An excess of NETs promote the production of proinflammatory cytokines in SARS-CoV-2 pulmonary disease, leading to cytokine storm and, in consequence, to diffuse alveolar damage [5]. Dysregulated NETs formation in severe COVID-19 is responsible for the immunothrombosis of poor prognostic significance [5]. Zuo et al. found a strong correlation between neutrophil-activation markers/NETs and D-dimer (fibrin degradation product) in patients with thrombotic complications of COVID-19 [8]. The above-mentioned discoveries point to the fact that NETs are key pathogenic mechanisms in COVID-19 [8]. Of note are the findings that NETs production is associated with various other disseminated infections including sepsis. NETs are an important structure preventing the dissemination of microorganisms [2,5]. On the other hand, overproduction and persistence of NETs may activate an immune response that is destructive to the host tissues [2,5]. In the course of sepsis, NETs production is also triggered, by various pro-inflammatory mediators and activated cells: platelets, endothelial cells, tumor necrosis fact alpha (TNF-α), interleukin-8 (IL-8), nitric oxide and various autoantibodies [5]. NETs components, in particular histones, DNA fibers and antimicrobial proteins significantly contribute to lethality in sepsis [5]. All these associations between NETs and sepsis have been described in detail by Gierlikowska and Demkow [3].

### 2.4. NETs in Cancer

NETs emerged as important players in contributing to tumor growth and metastasis formation—all these processes are described in detail by Demkow [4]. NETs have the ability to modulate the evasion capacities of the tumor cells [4,18]. To summarize, NETs awaken dormant cancer cells, promote cancer cell extravasation, enhance proliferation and migration of cancer and regulate the tumor microenvironment by degrading the extracellular matrix through the secretion of proteases providing a niche for metastatic tumor [4]. Moreover, NETs initiate the mesenchymal transition of the epithelial cells and potentiate migratory and invasive abilities of cancer cells. Circulating tumor cells, when entrapped by NETs fibers, can be sequestered and brought to distant organs forming lymphatic or hematogenous metastases [4,18]. Among the NET-driven tumorigenic activities, NETs directly affect the characteristics of tumor cells through activating signals, thus enhancing the invasiveness of cancer cells [4]. Furthermore, as mentioned above, NETs fuel cancer-associated thrombosis. Finally, NETs surround the primary tumor forming a barrier blocking the access of cytotoxic T cells and natural killer cells, thereby facilitating immune escape from the immunosurveillance [4]. The latter effect is not opposing the previously described mechanisms as it is responsible for the development of an immunosuppressive microenvironment fueling tumor growth, thus allowing progression and metastasis.

### 2.5. NETs in Ischemic Stroke

It has been widely recognized that NETs can contribute to the pathogenic mechanism of various diseases affecting the central nervous system, such as ischemic stroke or systemic sclerosis as currently described by Manda-Handzlik and Demkow [6]. Ischemic stroke is usually caused by local thrombosis in the brain circulation or migration of peripheral clot responsible for vascular occlusion blocking the oxygen supply of the brain. NETs further promote secondary thrombosis, extending the period of ischemia. It is also postulated that the no-reflow phenomenon, impairing t-PA-induced thrombolysis, may be attributed to the NETs conglomerates entrapping platelets and activating intrinsic coagulation pathway in the brain capillaries [6,19,20].

## 3. The Mechanisms of NETs Degradation

### 3.1. DNA Degrading Enzymes

The NETs clearance is necessary to maintain the correct balance between NETs formation and degradation [2,4]. The effective removal of extracellular DNA is crucial for tissue homeostasis, the prevention of inflammation and to avoid the presentation of auto-antigens [2]. Although many researchers have been exploring the process of NETs generation and pathophysiology, the knowledge on their degradation and the restitution of NETs-injured tissues is scarce [4,5]. Haider et al. suggested that NETs are cleaved by a concerted action of extracellular and secreted DNases followed by intracellular degradation by macrophages [21]. The cleavage with DNases plays a major role among physiological processes maintaining a low concentration of circulating free DNA. As DNA is the main component of NETs, DNases emerged as fundamental enzymes that breakdown NETs in vivo [22]. The extracellular DNases hydrolyzing circulating DNA comprise of the two families: DNase I, DNase II, exhibiting slightly different biochemical properties but partially redundant roles. DNases hydrolyze phosphodiester bonds of DNA molecules. The primary evolutionary role of DNases is suggested to degrade bacterial DNA [23]. The DNase I family consists of four members: DNase I, DNase1L1, DNase1L2 and DNase1L3, while the DNase II family includes DNase II α, DNase II β and L-DNase II [24]. The ability to hydrolyze DNA is common for both families. DNases are expressed across multiple tissues [24]. The degradation of DNA by DNase1 and DNases1L3 is the rate-limiting factor for NETs accumulation. DNase1 and DNase1L3 cleare NETs in blood vessels in the course of sepsis or sterile neutrophilia [24]. All except one are encoded by *DNase I* and *DNase II*, while the putative gene coding L-DNase II is *SERPINB1* [24].

DNase I, mainly produced by the pancreas and kidneys, is the major nuclease present in the blood and other body fluids that cleaves extracellular dsDNA into fragments with 30-hydroxy and 50-phospho ends [24]. The structure and sequence of the DNA substrate affects the kinetics of hydrolysis—DNase I cleaves double-stranded DNA (dsDNA) 100–500 times faster than single-stranded DNA (ssDNA) [24].

DNase II digest phosphodiester backbone of DNA resulting in the formation of two fragments with 30-phospho and 50-hydroxy ends. This enzyme has the highest activity in the absence of divalent cations and at acid pH. DNase II resides in lysosomes of various cells including macrophages, and in multiple tissues, pointing to the role of this enzyme in the hydrolysis of phagocytosed fragments of exogenous DNA, mainly derived from apoptotic cells [25]. Nagata and coworkers have confirmed that DNAses play an important role during apoptosis and its deficiency activates innate immune response [26]. The same group found that DNase II-deficient mice develop polyarthritis attributable to an overproduction of TNF-α by macrophages accumulating undigested DNA [27]. Conversely, Ferrera et al. did not observe excessive accumulation of NETs-derived DNA in macrophages nor TNF-α release as a result of DNase II silencing. Moreover, these authors claim that DNase II plays a role in the detection of NETs-derived DNA in cells costimulated via TLRs [28].

The evidence has accumulated that, apart from DNase family, there are other enzymes dismantling the NETs structure such as 3′-exonucleases (TREX1 and TREX2) [24]. TREX1 cleaves DNA fragments remaining in the course of DNA replication, apoptosis, netosis, DNA repair and recombination pathways. The 3′ to 5′ exonucleases-dependent DNA fragmentation results in the release of DNA 3′ termini necessary for downstream events critical for DNA repair or replication, i.e., the excision of modified, mismatched, fragmented, damaged or even normal nucleotides [24]. What is more, the 3′ to 5′ proofreading of DNA synthesis represent the most important mechanism securing genome stability. If 3′ exonuclease activity fails, the cell cycle defects, genome instability and enhanced radiation sensitivity results in mutagenic DNA changes promoting cancerogenesis [29,30]. A relevant role of TREX family in the process of NETs degradation is related to its potential to destroy oxidized DNA which is resistant to DNAses I and II. As oxidative stress is an important mechanism in the process of NETs formation, the oxidized form of DNA is largely present and exposed in NETs. Furthermore TREX1, activates the cGAS–STING intracellular pathway through a BAK/BAX-dependent process, leading to misbalance in type I interferon synthesis and immune dysregulation/autoimmunity [22,31]. The clinical consequences of TREX1 deficiency was described by Morita et al. in TREX1 null mice [30]. The knock-out mice presented interferon-dependent autoimmune response resulting in inflammatory myocarditis progressing into dilated cardiomyopathy with fatal consequences [30]. Mutations in the gene encoding TREX1 were further associated with common and rare autoimmune and inflammatory conditions [31]. The intracellular degradation of NETs by macrophages is also dependent on TREX1 function [32]. It was also demonstrated that dendritic cells degrade NETs using DNase1L3. In the light of these observations, it can be assumed that a concerted action of all mechanisms involved and extracellular DNA degradation is necessary to maintain the homeostasis of the immune system [32]. The effects of TREX1 and TREX2 are clearly distinguishable. TREX2 supports the genome integrity of keratinocytes playing a role in DNA damage removal and degradation of removed fragments [29]. Recent evidence strongly supports the opinion that TREX2 complex is involved in the transcription processes and nuclear messenger RNA transport in mammalian cells [33].

### 3.2. NETs Degradation by Macrophages

The work of Farrera et al. suggests that DNase I in physiological concentrations is not sufficient to completely degrade NETs, pointing to an additional mechanism necessary for the decomposition of this structure. These authors highlight a prominent role of macrophages in NETs degradation [28]. Macrophages and neutrophils are important cells of an innate immune response and act in cooperation. The interaction between polymorphonuclear cells (PMN) and macrophages has been suggested as a crucial mechanism modulating inflammation in the course of many pathological conditions [34]. Macrophages helps eliminate damaged cells and debris in their microenvironment. Moreover, macrophages scavenge foreign invaders or apoptotic/necrotic cells protecting the organism from potential danger signals [35]. They are strategically located and have the ability to uptake and process infectious agents and many other particles [34,35]. The macropinocytosis and endocytosis of DNA fragments assure a counterbalance of NETs generation and degradation, necessary for the maintenance of proper homeostasis [34,35]. Macrophages, being key modulators of extracellular DNA degradation, phagocytose NETs elements without giving rise to an inflammatory reaction, however, if preactivated with microbial products such as LPS, they secrete proinflammatory cytokines such as IL-1β, interleukin-6 (IL-6) and tumor necrosis factor-α (TNF-α) and begin effective antigen presentation [36]. Proinflammatory stimulation of macrophages and dendritic cells prime these cells for enhanced uptake and breakdown of NETs. [32,37].

The same directionality of effect was observed by Farrera et al. who confirms that macrophages are capable of the efficient clearance of NETs by taking up the extracellular DNA [28]. This process is facilitated by the extracellular digestion of large fragments of NETs by DNase I secreted by macrophages, as well as by the opsonization of NETs with complement factor 1q (C1q) [28]. The blocking of macropinocytosis in mice bearing a thrombus led to prolonged resolution of the clot; moreover, the NETs amount inside the thrombus was increased [28]. Preprocessing of NETs by DNase I and/or opsonization with C1q facilitated their clearance by macrophages in a cytochalasin D-dependent manner [28]. These authors have also shown that transfection of NETs or NET DNA inside macrophages stimulates the production of interferons, whereas the normal uptake of NETs by macrophages is immunologically silent, i.e., is not inducing the production of any mediators of the immune response [28]. The experiments with chloroquine proved that NETs undergo degradation in lysosomes, however the involvement of other cell compartments is not excluded [28]. Haider et al. aimed to determine the capacity of macrophages to degrade NETs and to identify the mechanism of endocytosis pathway as well as to investigate whether polarization of macrophages may change the kinetics of uptake and degradation [21]. These authors also provided evidence that local macrophage density in tissue sample from human aortic aneurysm is inversely associated with the presence of NETs in the tissue [21]. Haider et al. claim that an effective degradation of naked DNA or oligonucleotides, as well as the NET-degrading ability of polarized macrophages activated by proinflammatory stimuli (LPS + IFN-γ) is augmented. Long-term polarization by LPS and IFN-γ increased amounts of DNase1L3 and DNase 2 in macrophages [21]. These results are consistent with a previous observation by Farrera et al. showing that unpolarized macrophages using DNase1 were able to cut NETs into smaller fragments [28]. These authors already demonstrated that the knockdown of DNase 2 does not inhibit NETs degradation by unpolarized macrophages and that cytoplasmic TREX1 activity is needed for effective NETs breakdown by macrophages. [28] Haider et al. also claim that degradation of NETs by both unpolarized and polarized macrophages, as well as by their respective conditioned media, is abrogated by an inhibitor of DNase activity—EDTA [21]. Haider et al. further characterized the repertoire of DNases in human unpolarized and polarized macrophages and showed that the main secreted form of deoxyribonuclease in macrophages is DNase 1L3 [21]. Another discovery of this group led to a conclusion that DNase IL-1 intensively localized in the filopodia of activated macrophages. This might, at least partially, explain an increase in NETs degradation ability of proinflammatory macrophages [21]. Moreover, Haider and coworkers suggest that the intensive degradation of intracellular DNA is only partly due to the enhanced production of DNase. Another complementary process required for the effective clearance of NETs in the thrombi is macropinocytosis [21]. The experiments on murine thrombosis model using Sytox green-labeled DNA provided further evidence that preventing micropinocytosis by imipramine (a selective inhibitor of micropinocytosis) increased the presence of NETs components in the thrombi found in vasculature and concomitantly decreased fibrinolysis, supporting the statement that macropinocytosis is an important mechanism playing a role in the uptake of NETs by macrophages, both in vitro as well as in vivo [21]. The inhibition of macropinocytosis by imipramine resulted in longer and wider thrombi with increased NETs content [21]. The blockade of phagocytosis through inhibiting actin polymerization or phagosome-lysosome fusion also reduced NETs breakdown [21]. Previous observations suggested a potential mechanism of an activation of macropinocytosis dependent on the stimulation of Toll-like receptors in the macrophages [38]. In addition, macropinocytosis has been described to be altered in differently polarized macrophages [39]. Additionally, Haider et al. continued their experiments on human model using samples from aortic aneurysm patients who underwent surgery [21]. Of note was the finding that NETs are involved in the pathogenesis of aneurysms (inflammation, infiltration with macrophages, destruction of vascular wall and formation of trombus) and are thus prominently found in human arterial tissue [21]. Haider et al. showed that local macrophage density in human aortic aneurysms was negatively associated with surrounding NETs in the intraluminal thrombi as well as in the vessel wall [21].

Li et al. further explored the role of macrophages in NETs degradation process using the model of hepatocellular carcinoma (HCC) [40]. They found that diabetes-induced NETosis boosted HCC invasion in a NETs DNA-dependent manner. They confirmed that deficient DNASE1L3 expression in tumor tissues is a key cause responsible for the impairment of NETs DNA removal [40]. The resulting accumulation of NETs cause DNA-primed HCC cells to invade by activating the cGAS-ncNF-κB signaling pathway [40]. These observations were further confirmed by Wang et al. who showed that expression of DNASE1L3 is very low in HCC tissues, which may create a NETs DNA-rich microenvironment, thereby promoting cancer invasion and/or metastasis [41].

All the above-mentioned mechanisms of NETs degradation are presented in Figure 1.

## 4. NETs Degradation Defects

The effective clearance of NETs prevents overactivation of the immune system with concomitant thrombosis [2,5]. The inefficient dismantling of NETs may potentially serve as a source of immunogens derived from these structures, i.e., DNA, histones, enzymes and other NETs components [6,11,12,13]. Recently, different studies highlighted the link between NETs clearance defects and clinically relevant autoimmune disorders, especially SLE and vasculitis [6,22]. Overall, DNase activity is required to prevent the spontaneous formation of intravascular thrombi containing NETs [14]. All mechanisms involved in NETs degradation, as described above, can be impaired. Low DNase activity and functional impairment can be caused by the generation of anti-DNase inhibitors (and/or anti-DNAse autoantibodies) or mutations occurring in DNAses genes [42]. Genetic mutations affecting DNASE1, DNASE2, DNASE1IL3 and TREX were described [24]. DNase1 and DNase1-like 3 are independently expressed and thus provide dual host protection against the deleterious effects of intravascular NETs [24]. In vivo studies using *DNASE*-knocked-out mice confirmed the direct correlation between DNase activity and autoimmune diseases [43]. Knockout mice lacking both deoxyribonucleases rapidly died from multiorgan failure due to rapid occlusion of blood vessels with NETs containing clots [43]. In patients with severe bacterial infections, vascular occlusions were invoked by a defect in NETs removal ex vivo manifested as the formation of intravascular NETs—bearing thrombi [7]. Another mechanism that may lead to DNase functional impairment is the presence of circulating DNase inhibitors or the generation of anti-DNase antibodies [42].

### 4.1. NETs Degradation Defects in SLE

A decade ago, Hakkim et al. first focused on the central role of DNase I for disassembling NETs, and then correlated the functional defects of DNase I with the impaired degradation of NETs in a subset of patients with SLE [22]. They further showed that, in selected patients named as ‘non-degraders’, a balance between NETs production and degradation was restored by the sera of healthy donors or discarding antibodies from SLE patients serum [22]. In the light of these observations, the authors hypothesized that the presence of anti-DNase I antibodies or DNases I inhibitors in the sera of SLE patients is responsible for the disease flares and kidney involvement [22]. A strong association between the reduction of DNases activity and the accumulation of NETs in autoimmune conditions was reported [22]. Insufficient production of DNase I (mutations occurring in DNase1 and DNase1L3 genes) or a decrease of its activity (DNase inhibitors or the generation of anti-DNase antibodies preventing the enzyme access to NETs) result in an inefficient degradation of free-circulating DNA and could determine the production of anti-nuclear autoantibodies (ANA) associated with SLE and LN being both a biomarker and a pathogenic factor contributing to the development of this condition [22,44].

The inverse correlation between circulating DNase1L3 and the formation of antichromatin and anti-dsDNA antibodies, with clinically relevant SLE-like disease and renal involvement, was also confirmed in animal studies [45]. *DNASE1L3*-deficient mice develop a typical lupus syndrome and have been widely used to support a direct implication of DNASE 1L3 in SLE/LN [45]. Yasumoto et al. presented two cases of patients with SLE and autoimmune glomerulonephritis bearing stop codon mutations in exon 2 of *DNASE1* [46]. The patients with genetic deletion of DNase I had high levels of anti-DNA antibodies and low levels of circulating DNase I, as well as IgG and (complement factor 3) C3 glomerular deposition [42]. In LN, the removal of DNA, and consequently of NETs, may be impaired for different reasons, including key actionable mutations in genes encoding the DNases [42,45].

A second mechanism that may lead to DNase functional impairment is the presence of DNase inhibitors in the sera of patients with low DNase activity [45], or the generation of anti-DNase antibodies [42]. The loss-of-function mutations in genes encoding nucleases is considered as an important mechanism determining the development of autoimmunity [42]. DNase I-knocked out mice presented with typical symptoms of SLE, including presence of ANA, aggregation of immune complexes in kidneys, development of glomerulonephritis and further death [44,47].

Congruently, a causal relationship in human studies between mutations in DNAse I are linked to SLE, and a direct correlation between low activity of DNase I and SLE is confirmed [48]. Low DNase I activity is implicated in multiple systemic and organ-specific autoimmune diseases including thyroid autoimmunity, Sjogren’s syndrome and severe inflammatory bowel diseases [49]. It has been appreciated that low DNase activity is both a biomarker and a pathogenic factor in SLE [24]. Hakkim et al. discovered that impaired ability to clear NETs by SLE patients may account for the pathogenesis of LN [22]. Both mechanisms were implicated: the presence of anti-NETs antibodies and DNase1 inhibitors. Impairment of DNase1 function and failure to dismantle NETs are correlated with kidney involvement [22]. The same directionality of effect was observed by Bruschi et al. who tested NETs profiles in SLE patients and discovered that circulating NETs markers increased in 216 SLE patients, half of which had incident LN [50]. These authors found a significant correlation between high NETs marker levels, high anti-dsDNA antibody levels or low C3 activity and the presence of LN associated with either high anti-dsDNA antibody-circulating levels or low C3 activity. DNase activity was found to be selectively decreased in patients with LN compared to patients with SLE without kidney involvement and to the healthy controls, despite similar serum levels of DNASE I [50]. More recently, Hartl et al. provided evidence for the direct implication of anti-DNase antibodies in the pathogenesis of SLE in humans complicated by different organ pathologies [51]. They have also explored the mechanism of this association discovering that IgG autoantibodies to DNase 1L3 (but not to DNAse I) in serum are responsible for a decrease in enzyme activity in 50% of patients with LN as compared to patients with uncomplicated SLE or healthy controls [51]. In LN, DNase1L3 activity was also lower in patients with active proteinuria compared to those in remission. In accordance with the fact that DNASE 1L3 mutations are rare and could not account for the diminished DNase1L3 activity in 50% of the patients, an autoimmune mechanism was suggested [51]. These scientists tested the ability of autoantibodies to DNase 1L3 to lower the activity of the enzyme and found that the high and specific binding of IgG to DNase 1L3 in the serum of patients with LN correlated with diseases activity [51]. Consistently, no binding to DNase I was observed [51]. Overall, the findings by Hartl et al. support the statement that anti-DNase 1L3 antibodies are responsible for the inhibition of this enzyme activity in patients with LN [51].

### 4.2. TREX Defects

The TREX1 disease-causing alterations include mutations and SNPs, and cause varied TREX1 dysfunction that might play a previously unanticipated role explaining the multiple clinical symptoms resulting from persistent oxidized DNA, as mentioned above, leading to enhanced type I interferon synthesis and immune dysregulation [22,31]. This mechanism links TREX1 deficiency with persistent NETs—dependent inflammation and autoimmunity [31]. Loss of function mutations in TREX1, both inherited and de novo, cause a spectrum of nucleic acid-mediated immune activation disease, including Aicardi–Goutieres syndrome, familial chilblain lupus and retinal vasculopathy with cerebral leukodystrophy and SLE [31,52]. These genetic discoveries have established a causal relationship between TREX1 mutation and autoimmune diseases [53].

### 4.3. Other Clinical Consequences of DNase Mutations

Overall, deletions or mutations of any *DNASEs*, although rare or ultrarare, are always associated with a chronic inflammatory condition accompanied by the autoimmune glomerulonephritis [42,54,55]. Leffler et al. described three children with homozygous mutations in *DNASE2 associated with* a decreased degradation of NETs [55]. All three patients had similar clinical phenotype: membranoproliferative glomerulonephritis, fibrosing hepatitis and recurrent fever [55]. None of the patients fulfilled the clinical criteria of SLE and the serum levels of anti-DNA antibodies were variable [55]. All cases were compatible with an IFN-mediated inflammatory disease that also characterized SLE [55]. The pediatric onset of monogenic familial SLE with glomerulonephritis and very high anti-dsDNA antibodies is evoked by mutations of *DNASEIL3* [55]. Additionally, these conditions can be manifested as urticarial vasculitis syndrome and hypocomplementemia, further progressing to severe SLE [56]. As another example of polymorphic changes in *DNASE1L3 (rs35677470),* it was linked to the family of autoimmune connective tissue diseases such as scleroderma, SLE or rheumatoid arthritis [57]. All these antoimmune conditions are present with functional defects of NETs degradation. Persistent NETs start a cascade of adaptive immune responses and complement activation and the deposition of NET-specific autoantibodies, creating a vicious circle of failed degradation and immune stimulation directly implicated in the pathogenesis of SLE [42,54,55,56]. Similarly, the presence of anti-DNase antibodies produced in the response to persistent NETs was described to be associated with microscopic polyangiitis (MPA) [58,59]. MPA patients had decreased DNase I activity in sera. Both IgG depletion from myeloperoxidase-ANCA (MPO-ANCA)-associated MPA sera and the supplementation of DNase I synergistically restored NET degradation [59].

## 5. The Potential Applications of NETs-Inhibiting Drugs

This review focuses on the NETs—degrading mechanisms, suggesting a new way to design novel therapeutics for the management of a diverse set of NETs-dependent indications. The above-mentioned observations support the statement that the digestion of extracellular nucleoproteins may have a significant potential for the prevention and treatment of PMN-mediated disorders, including autoimmune diseases, exaggerated inflammatory reactions, severe infections and cancer. Further investigations on the inhibition of NETosis pathway as well as NETs degrading drugs provide potential therapeutic avenues for autoimmune diseases, especially SLE. An interesting option is also the combination of classical and anti-NETs intervention. A recent review by Mutua and Gershwin summarized the current knowledge on potential anti-NETs therapeutics [60].

Certain widely applied SLE therapeutics, such as tacrolimus, cyclosporine A and chloroquine, are targeting NETs components or interfering with mechanisms of NETs formation [61]. With the recent advances in the knowledge of how to inhibit or degrade NETs, several approaches to develop strategies to NET-targeting can be considered. DNAses are the most important enzymes dismantling NETs DNA. Gupta and Kaplan demonstrated that the administration of DNase 1 diminished SLE activity in mice [62]. They showed that TAK-242, a TLR4 inhibitor, decreased NETs formation, suggesting a therapeutic effect on autoimmune diseases [62]. In addition, PF1355, an inhibitor of MPO, limits the progression of autoimmune vasculitis in mice [62]. The modulation of either the NET production or the DNA removal appears as two possible effective strategies in SLE/LN treatment, and a balance of the two approaches may produce a synergy. On the other hand, blocking NET production may fail and, in some cases, may negatively impact the general clinical status and severe infectious complications. Blocking NET production is still an experimental area of investigation and further studies are warranted to explore this therapeutic option [50]. According to Pagnoux et al., the increase of DNase due to removing or blocking the synthesis of the circulating autoantibodies decreases the concentrations of circulating chromatin in SLE patients and propose plasmapheresis to decrease autoantibody levels [63]. Therapeutic plasma exchange has been widely used in many autoimmune disorders; however, further studies are needed to confirm its efficacy in NETs-dependent conditions [63]. On the basis of the reviewed studies, we may suggest that the blockade or the selective depletion of anti-DNase autoantibodies, or other strategies aimed at reducing NETs formation, could create a potential therapeutic option to prevent the progression of SLE and LN. Novel approaches to correct NETs-related tissue damage focused on the use of a recombinant human DNase-1 (dornase alpha—mucolytic agent applied in cystic fibrosis). De Buhr et al. observed the ability of DNase to degrade NETs in the lungs of calves infected with bovine respiratory syncytial virus [64]. Park et al. confirmed the effectiveness of DNase-1 coated nanospheres as modulators of NETs-associated complication of severe infection in mice [65]. Consistent observations were noted in SARS-CoV-2 patients [66]. The experimental use of DNase -1 coated melanine-like nanospheres on the plasma of COVID19 patients resulted in the significant reduction of NETs and MPO activity, as well as the decrease of the cytokines IL-1β, IL-6 and TNFα, involved in NETs vicious circle [66]. Nevertheless, the NETs remnants may be responsible for the development of bacterial superinfection in COVID-19 patients [5]. Thus potential benefits of DNase containing products in SARS-CoV-2 infection have to be confirmed by further studies. The other naturally occurring molecule, reducing pathological NETs activity is alpha-1-antitripsin (AAT), a neutrophil elastase inhibitor [67].

AAT binds extracellular IL-8, reducing the neutrophils’ influx to the inflammatory site and augments neutrophil superoxide production, inhibiting the activity of neutrophil elastase [67]. The other beneficial effects of AAT depend on the inhibition of endothelial cells apoptosis and thrombin generation [67]. These properties of the drug may be important in reducing the NETosis and immunothrombosis in the course of SARS-CoV-2 infection. Moreover, AAT expresses the natural anti-SARS-CoV-2 activity as inhibitors of S-protein cleavage [5]. During acute-phase reaction, especially in the course of severe infections, circulating AAT levels increase [67]. Moreover, the individuals diagnosed with AAT deficiency were more prone to the development of uncontrolled infections. Vianello et al., while looking for predictors of severe SARS-CoV-2 disease in Italian population, proved the geographic co-localization of AAT deficiency and SARS-CoV-2 infections [68]. Thus, it is possible that the patients with severe SARS-CoV-2 disease could benefit from therapeutic AAT administration [67]. It is also demonstrated that the direct inhibition of the process of NETosis can prevent COVID-19 exacerbation.

As another example, recombinant DNases may play a very important role as a potential drug in monogenic SLE. It is also demonstrated that DNase I digesting the NETs can destruct the scaffold of clot formation, suggesting the potential therapeutic role of the enzyme in the development of NETs-dependent thrombosis [23]. Gupta and Kaplan observed that calcineurin inhibitors blocking calcium mobilization required for NETosis (cyclosporine A and tacrolimus) are effective medications for SLE patients [62]. Furthermore, N-acetyl cysteine (NAC), a potent ROS scavenger, confers inhibiting effects of NETs extrusion because of its sharp reliance on ROS production, while exerting therapeutic effects in autoimmune diseases. NAC was effective in the therapy of SLE patients as confirmed by two clinical studies [62]. The evidence has also accumulated that Mito TEMPO, a specific inhibitor of ROS production, hindered NETosis and concomitantly decreased activity of SLE in mice. Moreover, the pharmacological inhibition of PAD activity attenuated the clinical course and reduced organ damage in the mice model of SLE and RA [62]. Similarly, again on the mice model, the inhibition of NET formation by Cl-amidine inhibited arterial thrombosis and diminished vascular damage [69]. Furumoto et al. described an inhibitory effect of tofacitinib on NETs, combined with an amelioration of vascular damage in the course of murine lupus [70]. Consistent with the antidiabetic drug metformin, inhibiting the NETs DNA-pDC-IFNα pathway reduced the risk of SLE exacerbations and corticosteroid dose in SLE patients [71]. Furthermore, Handono et al. observed the protective effect of vitamin D3 on NETs-dependent endothelial damage in SLE patients by blocking the externalization of neutrophil elastase [72]. Finally, anti-NETs therapy is believed to prevent the awakening of dormant cancer cells to inhibit the spreading of tumors as well as the formation of metastases [4].

## 6. Conclusions

The present review highlights complex interactions between the generation and degradation of NETs. Focusing on NETs degradation mechanisms may provide novel insights into the therapy of cancer, severe infections including COVID-19, or autoimmune diseases and many others [73]. The overproduction of NETs confirmed by high levels of circulating NETs markers or the presence of NETs in tissue samples may stand for the identification of patients who could benefit from NET-targeting therapy [74]. NETs degrading drugs may supplement other therapeutic regimes applied to prevent or treat cancer, autoimmunity and immunothrombosis. As mentioned above, numerous researchers have developed promising concepts on anti-NETs strategy [4,5,6]. The potential benefits of destroying NETs in vivo encourage further research. Several anti-NETs approaches had therapeutic effects on animal models of cancer and autoimmune diseases; nevertheless, the development of new drugs for patients needs further study and more time necessary for the effective development of clinical compounds able to target NETs [4,5,75,76]. Both options, either to dismantle formed NETs, or to block their production, require further study to enable clinicians to be more confident to use those drugs. Such strategies and underlying molecular mechanisms are at the preliminary phase and further data to explore their therapeutic potential and potential severe side-effects are highly anticipated. On the other hand, the risk of systemic infections in NETs-depleted patients may limit clinical applications in anti-NETs therapy and further study is warranted to investigate this issue. Targeting NETs is a worthwhile strategy in contemporary medicine that can be envisioned thanks to the ground-breaking discovery of Brinkmann et al. [1].

## Figures and Tables

**Figure 1 ijms-24-04896-f001:**
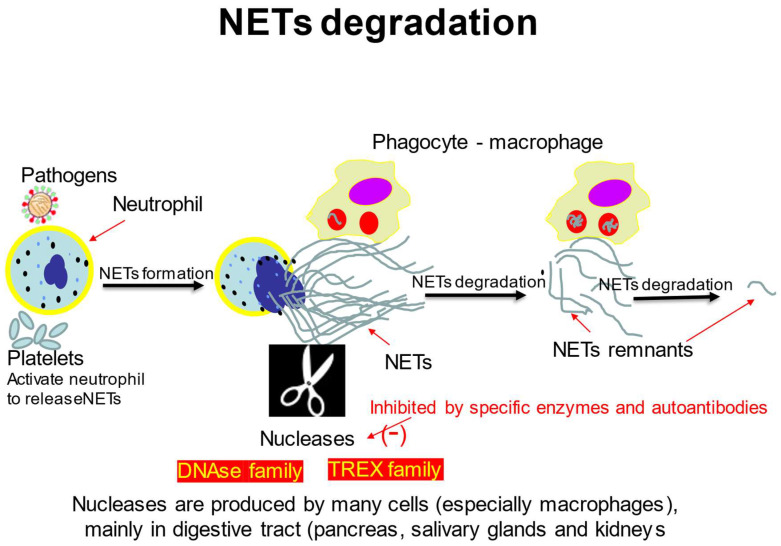
NETs degradation mechanisms.

## Data Availability

Not applicable.

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
