# Peer review of "Molecular Mechanisms of Neutrophil Extracellular Trap (NETs) Degradation"

_ijms, 2023, doi:10.3390/ijms24054896_

Round 1
Reviewer 1 Report (Previous Reviewer 2)
The authors have addressed almost all the comments from the previous round of review and have considerably improved the manuscript.
The only remaining concern in the text is Section 4.1., that was asked to be improved in the first round of revision by reducing redundancy of the text. In particular, similar information from ref 22 is mentioned (now in lines 400-403 and lines 440-442 of the manuscript version with red markings) and it is not clear what are separate messages of those paragraphs. This section will benefit from delineating more clearly which results are from mice and which are from humans.
A scheme of NETs degradation mechanisms is also added (Fig.1.). However, the format of the image doesn’t seem very appropriate for printing because of the black background. I would suggest to change the background color to white and correspondingly change color scheme of the figure to make it eye-catching.
The content of the figure has to be improved to match the quality of IJMS articles: 1) Platelets are depicted as very small crystals compared to a neutrophil. It is better if they will have cell-resembling shape and appropriate size, and also their function in NET formation has to be noted (or any other purpose of their presence in the figure).
2) The arrows that denote action or process (below “NETs formation” and “NETs degradation” ) should look different from the arrows used to sign image parts.
3) It is not clear from the figure, where do DNAses and TREX enzymes come from (which cells secrete those enzymes). Please add this information if it is known, or add a sign that at least macrophages release DNAses.
4) As an optional suggestion: it would be very good if the figure would depict other aspects of NET degradation regulation discussed in the manuscript: DNAse inhibitors, mutations, anti-DNAse antobodies
There are few typos that I noticed, however, I suggest careful editing throughout the manuscript before publishing as there could be other typos:
(Line numbers are for the manuscript version with red markings)
Line 352: “All” instead of “Ala”
Line 410: anti-nuclear antibodies (ANA) – add abbreviation, that is also used later
Line 421 : C3 appears for the first time, and explained only in line 446. Add “complement factor 3” (C3). Delete complement factor 3 from line 446.
418. “al.” instead of “all”
Author Response
I agree with all the comments of the reviewer 1. I will correct the manuscript and the figure accordingly
Reviewer 2 Report (Previous Reviewer 1)
The manuscript is a nice and deep overview on the mechanisms of NET degradation in physiology and pathology. The review is now well organized and easily readable. A little spell check is necessary, i.e. be sure that Covid19 is written always in the same manner (COVID19 or Covid19)
Author Response
I agree with all the comments of the reviewer 2. Small corrections of the english language accuracy will be introduced.
Reviewer 3 Report (New Reviewer)
Author had a great job for organizing and analyzing a lot of publications of neutrophil extracellular trap and its degradation in some field of clinic issues. I would like author to answer the following question before making an endorsement.
1. Is there any publication involved in NETs and/or NETs degradation in Sepsis? If yes, please have a short paragraph to description.
2. Since for neutrophil death, not only involved in NETs, but also can be through apoptosis, necrosis, necroptosis, pyroptosis and autophagy, how the different between NETs and others process during some clinic pathophysiology like sepsis and ARDS? Just have one of paragraph enough to write down the comparison in general.
Author Response
I agree with the reviewer comment. I will add some more information about the role of NETs in the course of sepsis and and some comment comparing the netosis and other neutrophil death mechanisms will be introduced.
This manuscript is a resubmission of an earlier submission. The following is a list of the peer review reports and author responses from that submission.
Round 1
Reviewer 1 Report
In this Review the Author addresses the role of Net degradation in the pathogens is of many diseases. The topic is interesting and of scientific soundness, however I have some concerns about its organization. In the present form the paper is heavy to read and mindless in. I think that the Author should the topic of NET degradation and the relative molecular mechanisms for each pathology faced in the manuscript.
minor concerns:
in paragraph 2, I do not agree with the sentence according to which stroke, atherosclerosis etc are not inflammatory disease. Inflammation has a very importante role in those pathologies. Please amend or delete.
in paragraph 3, it is not clear the role of citrullination in NET formation. Please explain
the entire manuscript should be revised by an English mother young.
Author Response
Thank you for the reviewer comments. Below I do address all theconcerns raised by the reviwer.
- Review the Author addresses the role of Net degradation in the pathogens is of many diseases. The topic is interesting and of scientific soundness, however I have some concerns about its organization. In the present form the paper is heavy to read and mindless in. I think that the Author should the topic of NET degradation and the relative molecular mechanisms for each pathology faced in the manuscript. I do corrected the latest version of the menuscript, making it more friendly to be read. All chapters were renomurred and additiond figure was added, which can help to follow the text.
minor concerns:
in paragraph 2, I do not agree with the sentence according to which stroke, atherosclerosis etc are not inflammatory disease. Inflammation has a very importante role in those pathologies. Please amend or delete. Amended
in paragraph 3, it is not clear the role of citrullination in NET formation. Please explain. Explained in both chapters describing the mechanisms of citrullination.
the entire manuscript should be revised by an English mother young. English language accuracy was revised and corrected by native speaker.
Reviewer 2 Report
This review article by U.Demkow is a profound survey of the role of neutrophil extracellular traps NETs in health and disease and, in particular, on the role of NETs degradation. It outlines the mechanisms of NETs degradation, implication of degradation defects in various diseases and potential treatments using anti-NETs approaches. It cites most recent original research papers on the topic as well as recent reviews relevant to each subsection. This review will be of interest to a broad readership, from immunologists to researchers and drug developers specializing in autoimmune diseases, infectious diseases including COVID-19, thrombosis and cancer.
The text of the paper is clear and well written with only minor inconsistencies and typos. I would suggest the following improvements:
1) A Figure depicting schematically the main players of NETs degradation process in various models and diseases, and highlighting which of them are to be targeted or substituted in anti-NETs therapies. Such a scheme would greatly help in understanding and remembering the key facts.
2) Most statements in subsections 1-7 are supported by citing review articles. It would be helpful if it is written explicitly that reader can refer to a cited review on this topic. And for any specific information it would be better if an original research is cited.
3) Some sentences are almost repeating each other. Combining them to eliminate this redundancy will make text more concise and clear. Examples: lines 356-359 and lines 382-385; lines 146 and 155.
4) Section 14 needs to be divided in several paragraphs conveying different drug applications, f.e. one starting from line 484 (about AAT), another from line 499.
Section 11 would also benefit from paragraph split.
5) Abstract: Line 16 "The ability to hydrolyze DNA by DNase I and DNAse II limits NETs accumulation". It is not clear if the author means ability of macrophages or something else.
Line 14: “..leading to the 13 development of various systemic and local injuries.” In this context the word “damages” would better convey the correct meaning than the word “injuries” because an injury is a damage caused by immediate physical stress, and presence of DNA doesn’t cause an immediate physical stress.
6) Section 12. What TREX defects have to do with NETs? Please add information about this connection, specifically if refs 52, 53 give this information.
7) Cite also a recent review: Mutua, V., Gershwin, L.J. A Review of Neutrophil Extracellular Traps (NETs) in Disease: Potential Anti-NETs Therapeutics. Clinic Rev Allerg Immunol 61, 194–211 (2021). https://doi.org/10.1007/s12016-020-08804-7
8) What is the definition of "immunologically silent " (line 268)? Please add an explanation to the text.
9) Minor comments:
- In section 6 it is stated that NETs promote cancer invasion and in the same time acts as a barrier blocking immune cells migration (lines 153 and 159). Is there any explanation of those opposing effects in literature?
- Numbering of section is not consistent throughout the paper. F.e. sections 3-7 fit better as subsections of section 2. Section 9 is supposed to be 8.2. Sections 11,12, 13 are supposed to be numbered as subsections of section 10.
- abbreviations have to be explained at the first occurrence, f.e. ANA , PAD, MPO, PMN
- similarly, mentioning chemicals and proteins names require introducing them at the first occurrence f.e. C1q at line 261 (compare line 264)
- some abbreviated terms (f.e. SLE, LN) are sometimes used without abbreviation later in the text. Please make it consistent using abbreviations or full term everywhere.
- Missing periods after some sentences.
Author Response
Thank you very much for very careful revision of my manuscript. I do agree withh all comment and I did corrected all the mistakes.
The text of the paper is clear and well written with only minor inconsistencies and typos. I would suggest the following improvements: Typos corrected
1) A Figure depicting schematically the main players of NETs degradation process in various models and diseases, and highlighting which of them are to be targeted or substituted in anti-NETs therapies. Such a scheme would greatly help in understanding and remembering the key facts. The figure is introduced
2) Most statements in subsections 1-7 are supported by citing review articles. It would be helpful if it is written explicitly that reader can refer to a cited review on this topic. And for any specific information it would be better if an original research is cited. I did refer to the cited reviews. All papers written by myself, contain the detailed explanation of the mentionned problem or mechanism.
3) Some sentences are almost repeating each other. Combining them to eliminate this redundancy will make text more concise and clear. Examples: lines 356-359 and lines 382-385; lines 146 and 155. Correcded avoiding redundancy.
4) Section 14 needs to be divided in several paragraphs conveying different drug applications, f.e. one starting from line 484 (about AAT), another from line 499. Done
Section 11 would also benefit from paragraph split. Done
5) Abstract: Line 16 "The ability to hydrolyze DNA by DNase I and DNAse II limits NETs accumulation". It is not clear if the author means ability of macrophages or something else. Explained
Line 14: “..leading to the 13 development of various systemic and local injuries.” In this context the word “damages” would better convey the correct meaning than the word “injuries” because an injury is a damage caused by immediate physical stress, and presence of DNA doesn’t cause an immediate physical stress. Corrected
6) Section 12. What TREX defects have to do with NETs? Please add information about this connection, specifically if refs 52, 53 give this information. Explained
7) Cite also a recent review: Mutua, V., Gershwin, L.J. A Review of Neutrophil Extracellular Traps (NETs) in Disease: Potential Anti-NETs Therapeutics. Clinic Rev Allerg Immunol 61, 194–211 (2021). https://doi.org/10.1007/s12016-020-08804-7 Cited
8) What is the definition of "immunologically silent " (line 268)? Please add an explanation to the text. Explained
9) Minor comments:
- In section 6 it is stated that NETs promote cancer invasion and in the same time acts as a barrier blocking immune cells migration (lines 153 and 159). Is there any explanation of those opposing effects in literature? Explained
- Numbering of section is not consistent throughout the paper. F.e. sections 3-7 fit better as subsections of section 2. Section 9 is supposed to be 8.2. Sections 11,12, 13 are supposed to be numbered as subsections of section 10. Done
- abbreviations have to be explained at the first occurrence, f.e. ANA , PAD, MPO, PMN Done
- similarly, mentioning chemicals and proteins names require introducing them at the first occurrence f.e. C1q at line 261 (compare line 264) Done
- some abbreviated terms (f.e. SLE, LN) are sometimes used without abbreviation later in the text. Please make it consistent using abbreviations or full term everywhere. Corrected
- Missing periods after some sentences. Added